# Projective Subspace Networks For Few-Shot Learning

## Abstract

Generalization from limited examples, usually studied under the umbrella of meta-learning, equips learning techniques with the ability to adapt quickly in dynamical environments and proves to be an essential aspect of lifelong learning. In this paper, we introduce the *Projective Subspace Networks (PSN)*, a deep learning paradigm that learns non-linear embeddings from limited supervision. In contrast to previous studies, the embedding in PSN deems samples of a given class to form an affine subspace. We will show that such modeling leads to robust solutions, yielding competitive results on supervised and semi-supervised few-shot classification. Moreover, our PSN approach has the ability of end-to-end learning. In contrast to previous works, our projective subspace can be thought of as a richer representation capturing higher-order information datapoints for modeling new concepts.

## 1 Introduction

Supervised learning with deep architectures, though achieving remarkable results in many areas, requires large amount of annotated data. Various studies show that many deep learning techniques in computer vision, speech recognition and natural language understanding, to name a few methods stated in Hinton et al. (2012); Krizhevsky et al. (2012), will fail to produce reliable models that generalize well if limited data annotations are available. Aside from the labor associated with annotating data, precise annotation can become ill-posed in some cases. One prime example of such a difficulty is object detection labeling which requires annotating bounding boxes of objects as explained in Alexe et al. (2012).

In contrast to the current trend in deep learning, humans can learn new concepts from only a few examples. This in turn provides humans with lifelong learning abilities. Inspired by such learning abilities, several approaches are developed to study learning from limited examples Lake et al. (2015); Lazaridou et al. (2017); Vinyals et al. (2016); Triantafillou et al. (2017); Xu et al. (2017); Finn et al. (2017); Ravi & Larochelle (2017); Wang et al. (2018); Mishra et al. (2018); Qiao et al. (2018); Neill & Buitelaar (2018). In machine learning, the diverse ideas in this context include embedding features through metric learning (Koch et al. (2015), Vinyals et al. (2016)), optimization technique(Finn et al. (2017), Ravi & Larochelle (2017)), and generative models(Fei-Fei et al. (2006), Lake et al. (2015)).

In this work, we propose a deep model that learns new concepts from limited data to address two challenging learning problems; namely: (**i**) few-shot classification and (**ii**) semi-supervised few-shot learning. The goal of few-shot classification is to learn a model that can discriminate a given query by comparing it to a few of samples (a.k.a. the support set). An example is to classify a motorcycle by viewing some different types of motorcycles (or other vehicles). The goal of semi-supervised few-shot learning is to additionally benefit from unlabeled data to boost the performance of the model.

Our method, coined Projective Subspace Networks or PSN for short, learns non-linear embeddings using subspaces, in a sense that samples of a class are modeled as a low-dimensional affine subspace. The use of subspaces to model images and sets has a long history in computer vision and machine learning. For example, it has been proved that the set of all reflectance functions (the mapping from surface normals to intensities) produced by Lambertian objects lie close to a low-dimensional linear subspace (Basri & Jacobs (2003)).

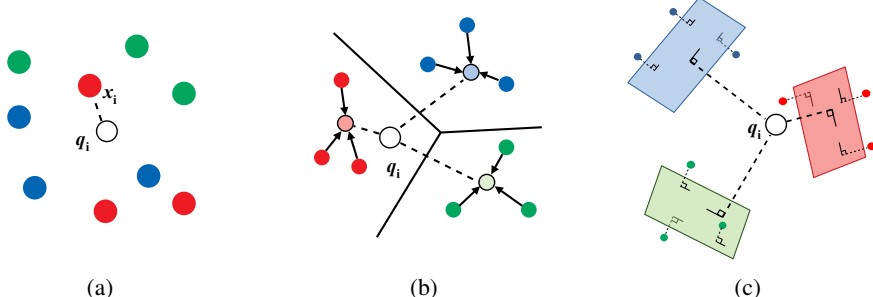

(a)                 (b)                 (c)

Figure 1: Feature embedding in (a) Matching Networks (Vinyals et al. (2016)), (b) Prototypical Networks(Snell et al. (2017)), and (c) our PSN method.

In spite of its intriguing properties, to the best of our knowledge, the subspace modeling has never been used previously to address few-shot learning problems. This makes our paper distinct and novel as compared to former studies (*e.g.*, Vinyals et al. (2016); Snell et al. (2017)). Fig. 1 provides a conceptual illustration of our approach.

We empirically observed that embeddings tailored towards capturing the structure of each class through low-dimensional affine subspaces could lead to discriminative models. Interestingly, such models can be built with minimum overheads and without opting for advanced methods in subspace creation (*e.g.*, such as the notion of sparsity). Our conjecture here is that subspaces are less sensitive to perturbations such as outliers and noise compared to other embedding techniques for few-shot learning as illustratively shown in Fig. 2.

Our contributions in this work are:

    i. Few-shot learning is formulated as an embedding problem through subspaces. We rely on a well-established concept stating that samples of a class (and hence variations such as pose and illumination) can be effectively captured by a low-dimensional affine space.

    ii. Adaptation from few-shot learning to semi-supervised learning is performed with a refinement through soft-assignment. The robustness of such a model is shown in our experiments.

    iii. We also introduce an evaluation mechanism to assess the generalization ability over unseen classes during test time.

## 2    RELATED WORK

In this section, we briefly review the literature on few-shot learning and subspace clustering. Few-shot learning was originally introduced to imitate human learning capabilities in classification. Some of the early works made use of generative models and similarity learning to capture the variation within parts and geometric configurations of objects (Fei-Fei et al. (2006); Lake et al. (2015); Torralba et al. (2007)). As of late, few-shot problems are mainly addressed through meta-learning techniques. For example, in Koch et al. (2015); Vinyals et al. (2016); Snell et al. (2017); Garcia & Bruna (2018), the problem of few-shot learning is formulated as non-linear embedding or representation learning.

The closest approach to our proposed work is the Prototypical Networks (PN hereafter) model ( Snell et al. (2017)). It uses the random choice of episode images which form the prototype center per class. In addition, recent metric learning approaches use complex architectures and pipelines in convolutional networks of Gidaris & Komodakis (2018) and Wang et al. (2018). In contrast, our projective subspace concept is extremely simple by design as it constitutes just a single layer of the network.

Another common meta-learning approach to few-shot classification uses meta-learning and manipulates gradient updates to train the model parameters. Ravi & Larochelle (2017) utilized Long-Short Term Memory (LSTM)-based learner to optimize the model parameters and their gradients. Moreover, the Model Agnostic Meta-Learner(MAML) proposed by Finn et al. (2017) used gradients per

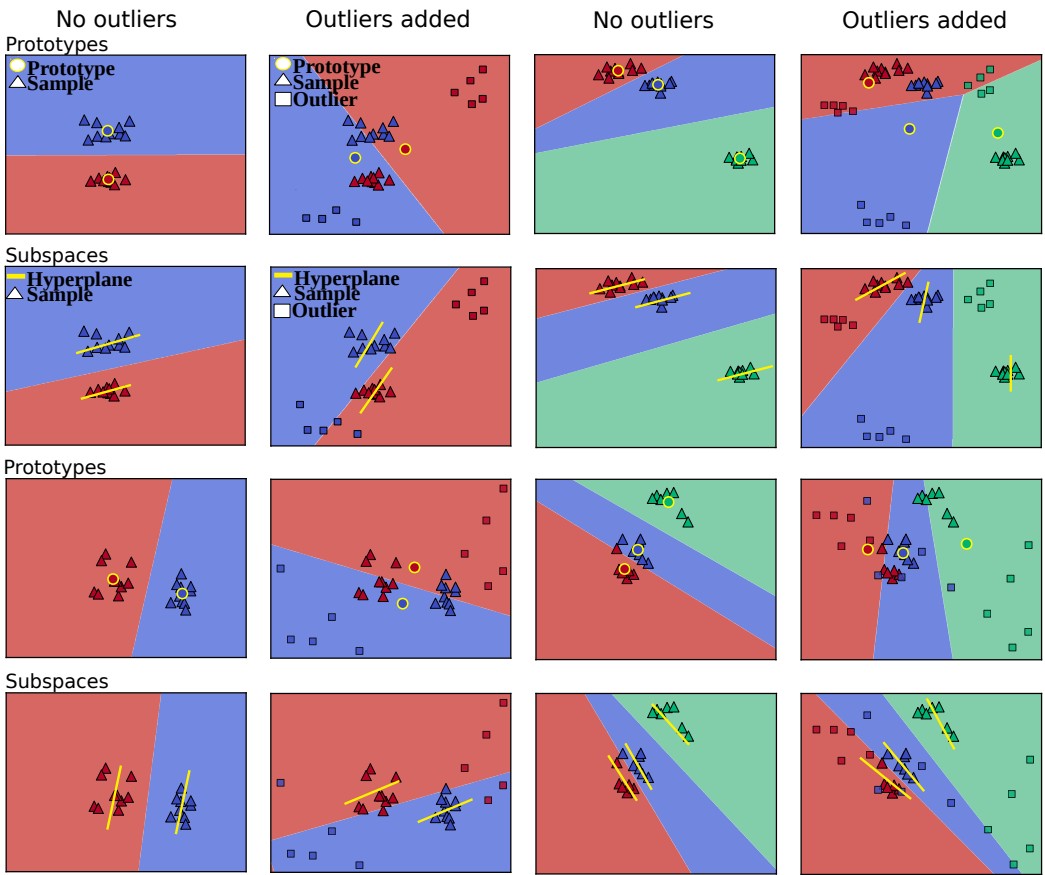

Figure 2: **The effect of outliers on prototypes and subspaces.** The odd rows show the decision boundaries obtained by prototypes (with and without outliers) for two- and three-class problems. The even rows depict how subspaces behave for the same problems. While affected, in general, subspaces show better resilience to perturbations and attain higher discriminatory power in comparison to prototypes. Best viewed in color.

task as well as a meta-gradient from combined tasks, and, as a result, it outperformed LSTM-based learner. Mishra et al. (2018) employed ResNets to model few-shot classification as a temporal solution with aggregation, thus, in meta-learning context, their model can refer to the past experience.

Our idea is based on the assumption that samples from any class in the support set form a low-dimensional subspace. As we will show, to train PSN, backpropagation through Singular Value Decomposition (SVD) is required. Backpropagation through matrix decomposition such as SVD is a well-studied problem Ionescu et al. (2015) with applications ranging from semantic segmentation to classification and visual recognition Ionescu et al. (2015); Li et al. (2017a); Gou et al. (2018).

## 3   PROBLEM SET-UP

We start by defining the terminology used in few-shot learning. The problem of $N$-way $K$-shot classification (*e.g.*, 5-way 1-shot) is defined as classifying queries belonging to $N$ classes by seeing only $K$ samples from each class. For example, in 5-way 1-shot setting, the model needs to identify a query among five classes by seeing only one sample from each class. To obtain a reliable model, so-called *episodes* are used in training. An episode $\mathcal{T}_i$ consists of two sets, the support set $S$ and the query set $Q$. The system is then trained by minimizing a classification loss over episodes, simulating the scenarios it will encounter at test time. This episode setting is the same as proposed by Vinyals et al. (2016).

A related problem is semi-supervised few-shot setting learning where unlabeled data is provided to the model. In the literature, various configurations are considered for semi-supervised few-shot learning (*e.g.*, Garcia & Bruna (2018); Boney & Ilin (2017); Ren et al. (2018)). In this work, we follow the challenging protocol in Ren et al. (2018) where so-called *distractors* are introduced. Here, an episode includes the support set $S$, query set $Q$, and unlabeled set $\mathcal{R}$. The support (labeled) $S$ and query $Q$ sets are configured as in few-shot learning. Additionally, an unlabeled set $\mathcal{R}$ is provided to assist the classification task within an episode. In the unlabeled set, there are examples from two different sources: the support classes and the *distractor* classes. As the name implies, examples from *distractor* classes are irrelevant to the classification task and represent classes outside the support set.

# 4 PROJECTIVE SUBSPACE NETWORKS

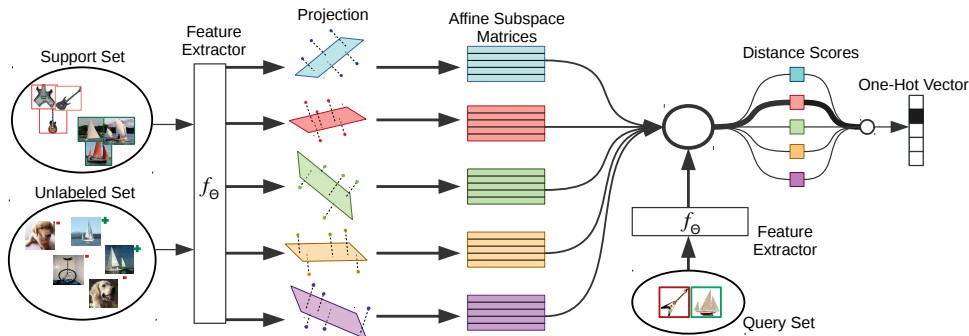

Figure 3: Projective Subspace Networks Architecture

In what follows, we introduce our PSN approach. The whole framework is trained end-to-end (see Fig. 3) which enables PSN to be used with various deep architectures. As an example, in § 5, we use PSN on top of the WideResNets (Zagoruyko & Komodakis (2016)) for few-shot classification.

We start by introducing our notations for the ($N$-way, $K$-shot) few-shot learning. Each episode or task $\mathcal{T}_i$ is composed of the support set $S = \{(\boldsymbol{x}_{1,1}, c_{1,1}), (\boldsymbol{x}_{1,2}, c_{1,2}), \cdots, (\boldsymbol{x}_{N,K}, c_{N,K})\}$ and the query set $Q = \{\boldsymbol{q}_1, \cdots, \boldsymbol{q}_{N \times M}\}$. Here, $\boldsymbol{x}_{i,j}$ denotes the $j$-th sample from class $i$ and $c_{i,j} \in \{1, \cdots, N\}$. In the semi-supervised setting, there is an unlabeled set $\mathcal{R} = \{\boldsymbol{r}_1, .., \boldsymbol{r}_U\}$ within an episode. We propose to model points by subspaces $\{\boldsymbol{Z}_i\}_{i=1}^N$. Each subspace $\boldsymbol{Z}_i$ has a basis represented by $\mathbb{R}^{D \times n} \ni \boldsymbol{P}_i = [\boldsymbol{p}_1, \cdots, \boldsymbol{p}_n]; n \leq D$, with $\boldsymbol{P}_i^\top \boldsymbol{P}_i = \boldsymbol{I}_n$.

## 4.1 PSN FOR FEW-SHOT CLASSIFICATION

Let $f_\Theta : \mathcal{X} \to \mathbb{R}^D$ be a mapping from the input space $\mathcal{X}$ to some $D$-dimensional representation realized by a neural network. Our goal is to learn $\Theta$, *i.e.*, the embedding function in a way that the resulting space is suitable for subspace representation. For simplicity, we assume that every class in an episode can be described by just one subspace. Extension to multiple subspaces per class is straightforward though. Define $\boldsymbol{\mu}_k$ as the mean of class $k$ in the embedded space. That is,

$$\boldsymbol{\mu}_k = \frac{1}{K} \sum_{i,\ c_i = k} f_\Theta(\boldsymbol{x}_i). \tag{1}$$

A basis for the subspace representing class $k$ can be obtained by Singular Value Decomposition (SVD). To be specific, we define $\boldsymbol{X}_k = [\boldsymbol{x}_{k,1} - \boldsymbol{\mu}_k, \cdots, \boldsymbol{x}_{k,K} - \boldsymbol{\mu}_k]$. Applying truncated SVD on $\boldsymbol{X}_k$ provides us with $\boldsymbol{P}_k$. We emphasize that more involved techniques to obtain robust subspaces from $\boldsymbol{X}_k$ can potentially improve the PSN. Nevertheless, our goal is to assess whether the concept of subspace modeling for few-shot learning is justified or not and thus we opt for truncated SVD in our implementation. Now a query $\boldsymbol{q}_j$ can be projected onto $\boldsymbol{P}_k$ which yields:

$$\boldsymbol{y}_{j,k} = \boldsymbol{P}_k^\top (f_\Theta(\boldsymbol{q}_j) - \boldsymbol{\mu}_k). \tag{2}$$

The distance from the query to $\boldsymbol{P}_k$ is:

$$d_{j,k} = \|f_\Theta(\boldsymbol{q}_j) - \boldsymbol{\mu}_k - \boldsymbol{P}_k \boldsymbol{y}_{j,k}\|. \tag{3}$$

We define the probability of the query assigned to class $k$ using a softmax function as:

$$p_\Theta(c = k|\boldsymbol{q}_j) = \frac{\exp\left(-d_{j,k}^2\right)}{\sum_{k'} \exp\left(-d_{j,k'}^2\right)}. \tag{4}$$

Now, we can minimize the negative log of Eqn. 4 to obtain $\Theta$. To train the whole framework, backpropagation through SVD is required which is available in modern deep learning packages such as PyTorch (Paszke et al. (2017)). Algorithm 1 explains the steps of training our PSN. The code will be released online on Github[1].

---

**Algorithm 1** Train Projective Subspace Networks

---

**Input:** Each episode $\mathcal{T}_i$ with $S = \{(\boldsymbol{x}_{1,1}, c_{1,1}), \cdots, (\boldsymbol{x}_{N,K}, c_{N,K})\}$ and $Q = \{\boldsymbol{q}_1, ..., \boldsymbol{q}_{N \times M}\}$

1:  $\Theta_0 \leftarrow$ random initialization
2:  **for** $t$ in $\{\mathcal{T}_1, ..., \mathcal{T}_{N_\mathcal{T}}\}$ **do**
3:      $\mathcal{L}_t \leftarrow 0$
4:      **for** $k$ in $\{1, ..., N\}$ **do**
5:          $\boldsymbol{X} \leftarrow \boldsymbol{S}_k$                                  ▷ Get examples in the support set from class $k$
6:          $\boldsymbol{\mu}_k \leftarrow \frac{1}{K}\sum_{\boldsymbol{x} \in \boldsymbol{X}} f_\Theta(\boldsymbol{x})$                        ▷ Mean from the support set
7:          $\boldsymbol{\mu}_k \leftarrow \dfrac{K\boldsymbol{\mu}_k + \sum_i m_i f_\Theta(\boldsymbol{r}_i)}{K + \sum_i m_i}$        ▷ Refined mean(only semi-supervised learning)
8:          $\tilde{\boldsymbol{X}} \leftarrow [\boldsymbol{x}_i - \boldsymbol{\mu}_k, ..., \boldsymbol{x}_K - \boldsymbol{\mu}_k]$
9:          $[\mathcal{U}, \Sigma, \mathcal{V}^\top] \leftarrow \text{SVD}(\tilde{\boldsymbol{X}})$                        ▷ Matrix factorization using SVD
10:         $\boldsymbol{P}_k \leftarrow \mathcal{U}_{1,...,n}$                                  ▷ Truncate the matrix
11:         **for** $\boldsymbol{q_j}$ in $Q_k$ **do**
12:             $\boldsymbol{y}_{j,k} \leftarrow \boldsymbol{P}_k^\top(f_\Theta(\boldsymbol{q_j}) - \boldsymbol{\mu}_k)$                        ▷ Query projection
13:             $d_{j,k} \leftarrow \|f_\Theta(\boldsymbol{q_j}) - \boldsymbol{\mu}_k - \boldsymbol{P}_k \boldsymbol{y}_{j,k}\|$                        ▷ Distance calculation
14:             $p_{j,k} \leftarrow \dfrac{\exp(-d_{j,k}^2)}{\sum_{k'} \exp(-d_{j,k'}^2)}$                        ▷ Softmax on distance scores
15:         **end for**
16:     **end for**
17:     $\mathcal{L}_t \leftarrow \frac{1}{N^2 M}\sum_k \sum_j -\log(p_{j,k})$
18:     Update $\Theta$ using $\nabla_\Theta \mathcal{L}_t$
19: **end for**

---

## 4.2 PSN FOR SEMI-SUPERVISED FEW-SHOT LEARNING

In what follows, we extend the model developed in § 4.1 to address semi-supervised few-shot learning. In doing so, we need to take advantage of the unlabeled data to fit better subspaces to our data. We achieve this by refining the center of each class according to

$$\tilde{\boldsymbol{\mu}}_k = \frac{K\boldsymbol{\mu}_k + \sum_i m_i f_\Theta(\boldsymbol{r_i})}{K + \sum_i m_i}, \quad \text{where} \quad m_i = \frac{\exp(-\|f_\Theta(\boldsymbol{r}_i) - \boldsymbol{\mu}_k\|^2)}{\sum_{k'} \exp(-\|f_\Theta(\boldsymbol{r}_i) - \boldsymbol{\mu}_{k'}\|^2)}. \tag{5}$$

Here, $m_i$ is the soft-assignment score for unlabeled samples. To work at the presence of *distractors*, we use a fake class with zero mean. We empirically observed that such a simple modification to the means can improve the results without the need of refining the SVD step.

## 5 EXPERIMENTS

Below we contrast and assess our method against state-of-the-art techniques on two challenging datasets, namely Mini-ImageNet (Ravi & Larochelle (2017)) and Tiered-ImageNet (Ren et al.

---

[1] https://github.com/anonymousauthor/PSN

| Models | 5-way 5-shot | 20-way 5-shot |
|---|---|---|
| Matching Nets (Vinyals et al. (2016)) | $55.31 \pm 0.73\%$ | $22.69 \pm 0.20\%$ |
| MAML (Finn et al. (2017)) | $63.11 \pm 0.92\%$ | $19.29 \pm 0.29\%$ |
| Meta-Learner LSTM(Ravi & Larochelle (2017)) | $60.60 \pm 0.71\%$ | $26.06 \pm 0.25\%$ |
| Meta-SGD (Li et al. (2017b)) | $64.03 \pm 0.94\%$ | $28.92 \pm 0.35\%$ |
| PN(Snell et al. (2017)) | $65.49 \pm 0.25\%$ | $37.23 \pm 0.21\%$ |
| **PSN** | $\mathbf{66.62 \pm 0.69\%}$ | $\mathbf{38.26 \pm 0.23\%}$ |

Table 1: Few-shot classification results on Mini-ImageNet using 4-convolutional stages with 95% confidence intervals.

(2018)). As the rule of thumb, the parameters of subspace dimension ($n$) that we used in the experiments are $K$-1 for training and two for testing stage.

**Mini-ImageNet.** The Mini-ImageNet(Ravi & Larochelle (2017)) contains 60,000 images of the ImageNet(Russakovsky et al. (2015)) datasets. Images in the Mini-ImageNet are of size $84 \times 84$ and represent 100 classes with 64, 16, and 20 classes used for training, validation, and testing, respectively.

**Tiered-ImageNet.** This dataset is also derived from ImageNet but contains a broader set of classes compared to the Mini-ImageNet. There are 351 classes from 20 different categories for training, 97 classes from 6 different categories for validation, and 160 classes from 8 different categories for testing. In contrast to the Mini-ImageNet, the training and test sets in Tiered-ImageNet represent distinct classes. Moreover, in designing the Tiered-ImageNet, the problem of few-shot learning with unlabeled data was taken into account and the labeled data is only within a small percentage when performing semi-supervised learning. In this large dataset, learning from the labeled data is still sufficient to produce reasonable representations even though the unlabeled data is set in huge portion *e.g.*, 90%.

## 5.1 FEW-SHOT LEARNING

We follow the general practice and evaluate our method on the Mini-ImageNet dataset when it comes to few-shot learning and classification. Various procedures such as pre-training using 64 classes (*e.g.*, Qiao et al. (2018)) and training with more classes/way have shown to increase the overall accuracy. Nevertheless, we avoid such procedures deliberately as we are mainly interested in contrasting the core idea, *i.e.*, the role of subspaces in few-shot learning. So, we trained on 5-way 5-shot and 20-way 5-shot, then applied the same classification task setup during testing. The CNN architecture is the same as the one used in Snell et al. (2017) with 4-convolutional stages. We also use WideResNets (Zagoruyko & Komodakis (2016)) with 16 depth, 6 widening factor, and 0.3 dropout rate to compare with ResNets (He et al. (2016)) solution reported by Mishra et al. (2018). The feature dimensions from both architectures are 1600 and 384 respectively. We used ADAM ( Kingma & Ba (2015)) for optimizing our model and set the learning rate to 0.001 and cut it to half every 2.5K episodes for both architectures.

**Results.** By design, the method cannot accommodate learning with exactly one example, hence, the comparison here is provided only for 5-shot in 5-way and 20-way. In every episode, the query set contains 15 samples from each class. Our method outperforms the previous methods with 4-convolutional stages shown in Table 1. We also implemented WideResNets (Zagoruyko & Komodakis (2016)) using our method and obtained the performance for 5-way 5-shot: $\mathbf{69.92 \pm 0.64\%}$ and 20-way 5-shot: $\mathbf{41.84 \pm 0.24\%}$ that can outperform ResNets-based approach proposed by Mishra et al. (2018) with $68.88 \pm 0.92\%$ in 5-way and 5-shot classification task.

## 5.2 SEMI-SUPERVISED FEW-SHOT LEARNING

In this experiment, the embedding architecture has 4-convolutional layers as PN (Snell et al. (2017)). We follow the experimental setup proposed by Ren et al. (2018). The episode composition for labeled or support set and query set is similar to the few-shot learning classification task, but there is an additional unlabeled set provided in each episode. Our model is trained on 100K episodes for

| Models | Mini-ImageNet 5-way 5-shot | | Tiered-ImageNet 5-way 5-shot | |
|---|---|---|---|---|
| PN, *Supervised* | $59.08 \pm 0.22\%$ | | $66.15 \pm 0.22\%$ | |
| **PSN,** *Supervised* | $\mathbf{63.43 \pm 0.61}\%$ | | $\mathbf{68.72 \pm 0.49}\%$ | |
| | **w/o** *Distractors* | **w/** *Distractors* | **w/o** *Distractors* | **w/** *Distractors* |
| PN-SSL, Non-Masked | $64.59 \pm 0.28\%$ | $63.55 \pm 0.28\%$ | $70.25 \pm 0.31\%$ | $68.32 \pm 0.22\%$ |
| PN-SSL, Masked | $64.39 \pm 0.24\%$ | $62.96 \pm 0.14\%$ | $69.88 \pm 0.20\%$ | $69.08 \pm 0.25\%$ |
| **PSN, Semi-Supervised** | $\mathbf{68.12 \pm 0.67}\%$ | $\mathbf{66.10 \pm 0.66}\%$ | $\mathbf{71.15 \pm 0.67}\%$ | $\mathbf{69.15 \pm 0.51}\%$ |

Table 2: Semi-supervised few-shot classification results on the Mini-ImageNet and Tiered-ImageNet with $40\%$ and $10\%$ labeled data, respectively. We show the classification results with and without *distractors*. We compare our results to PN on semi-supervised learning (PN-SSL) with soft $K$-means (non-masked) and masked $K$-means (masked), as proposed by Ren et al. (2018).

**Mini-ImageNet**

| Way($N$) | Shot($K$) | PSN | PN | Way($N$) | Shot($K$) | PSN | PN |
|---|---|---|---|---|---|---|---|
| | 3 | $\mathbf{61.26 \pm 0.79}\%$ | $60.36 \pm 0.31\%$ | | 3 | $\mathbf{45.46 \pm 0.44}\%$ | $44.09 \pm 0.28\%$ |
| | 5 | $\mathbf{66.62 \pm 0.69}\%$ | $65.49 \pm 0.25\%$ | | 5 | $\mathbf{51.49 \pm 0.46}\%$ | $49.06 \pm 0.25\%$ |
| 5 | 10 | $\mathbf{71.88 \pm 0.59}\%$ | $69.80 \pm 0.30\%$ | 10 | 10 | $\mathbf{57.14 \pm 0.41}\%$ | $55.23 \pm 0.27\%$ |
| | 15 | $\mathbf{73.50 \pm 0.58}\%$ | $71.92 \pm 0.26\%$ | | 15 | $\mathbf{60.20 \pm 0.39}\%$ | $58.06 \pm 0.22\%$ |
| | 20 | $\mathbf{74.88 \pm 0.63}\%$ | $73.01 \pm 0.31\%$ | | 20 | $\mathbf{61.38 \pm 0.44}\%$ | $59.27 \pm 0.22\%$ |
| | 3 | $\mathbf{37.07 \pm 0.33}\%$ | $36.24 \pm 0.22\%$ | | 3 | $\mathbf{32.13 \pm 0.25}\%$ | $31.43 \pm 0.18\%$ |
| | 5 | $\mathbf{43.18 \pm 0.32}\%$ | $40.94 \pm 0.24\%$ | | 5 | $\mathbf{37.71 \pm 0.23}\%$ | $35.58 \pm 0.17\%$ |
| 15 | 10 | $\mathbf{49.44 \pm 0.31}\%$ | $47.19 \pm 0.21\%$ | 20 | 10 | $\mathbf{44.09 \pm 0.22}\%$ | $41.69 \pm 0.16\%$ |
| | 15 | $\mathbf{51.96 \pm 0.32}\%$ | $49.85 \pm 0.25\%$ | | 15 | $\mathbf{47.27 \pm 0.22}\%$ | $44.37 \pm 0.17\%$ |
| | 20 | $\mathbf{54.23 \pm 0.34}\%$ | $51.37 \pm 0.20\%$ | | 20 | $\mathbf{49.00 \pm 0.23}\%$ | $45.90 \pm 0.18\%$ |

Table 3: Generalization performance of PSN and PN with varying number of way ($N$) and shot ($K$) at the test time.

Mini-ImageNet and Tiered-ImageNet with $40\%$ and $10\%$ of labeled data, respectively. We used the ADAM solver (Kingma & Ba (2015)), the set the learning rate to $0.001$ with the weight decay and cut the rate to half every 10K episodes. We trained in two settings: (i) *supervised* setting, where only labeled data is taken into account, and (ii) semi-supervised setting for which the unlabeled set is also used. The unlabeled set is composed of the examples from the classes in the support set and *distractor* classes. The number of supporting classes and *distractor* classes is set to five for training and testing. In the training stage, the number of examples in the unlabeled set is 50 consisting of five examples from each class. In the testing stage, the unlabeled set consists of 20 examples from each class. We also define the query set to have 20 examples per class for testing purpose.

**Results.** In these experimental results, the performance is counted over 600 episodes. The results are averaged over 10 random splits of labeled and unlabeled sets. The *supervised* experiment shows that our method learns robust feature embedding from a small portion of labeled data. With the help of soft-assignment over unlabeled datapoints, the *semi-supervised* experiment detailed in Table 2 is demonstrated to outperform Prototypical Networks for Semi-Supervised Learning (PN-SSL) proposed by Ren et al. (2018).

## 5.3 Generalization Beyond $N$-way $K$-shot

As a measure of generalization ability, we propose to evaluate algorithms beyond the somehow inflexible testing protocol of few-shot learning. In particular, we assess whether a model trained on low number of classes can generalize well to classification tasks involving large number of classes. To gain more insights, we further study how models trained with the $K$-shot assumption will perform if extra examples are available at the test time.

In the evaluation below, all models are trained with the 5-way 5-shot setting. At the test time, the models face the test protocol of $N$-way $K$-shot learning with $N \in \{5, 10, 15, 20\}$ and $K \in \{3, 5, 10, 15, 20\}$. Table 3 contrasts the performance of PSN against PN. The table is self-explanatory. In all experiments, PSN outperforms PN with the gap widened with more challenging settings (*e.g.*, 20-way 20-shot).

**Robustness to Perturbations.** Our motivation to develop PSN is to devise a model that shows better resilience to perturbations. Intuitively, to have a noticeable change in the orientation of a subspace, one needs to induce drastic changes to the set. To empirically verify our claim, we assess how robust is PSN in comparison to PN (Snell et al. (2017)) in the presence of perturbations. In particular, we considered the problems of 5-way 5-shot and 5-way 10-shot learning and introduced two types of perturbations at the test time to the trained models. Firstly, we randomly sampled examples from classes not presented in the support sets and included them in the support examples. Secondly, noisy examples are generated randomly using a multivariate Gaussian distribution with random mean and variance of $\sigma = \{0.15, 0.3, 0.4\}$. The results of this study are presented in Appendix A due to page limitations. To summarize, our experiments show that both PSN and PN are affected by outliers negatively. That said, the PSN exhibits a much better degree of resilience to outliers. When additive noise is considered, PSN behaves robustly for a wide-range of contamination. In contrast, the performance of PN drops rapidly and significantly in the presence of noise, reinforcing our idea that the use of subspaces indeed leads to a more robust model for the task in hand.

## 6  DISCUSSION

Conceptually, the PN Snell et al. (2017) is the closest work to ours as both solutions obtain a refined representation for each class in the support sets, with the former using the class mean as the representation while in PSN affine subspaces model classes. That said, PSN also uses the mean of each class towards identifying their representative subspaces. Aside from the theoretical properties of affine subspaces[2], we empirically observed that for more challenging setups (*e.g.*, 20-way 20-shot), utilizing the mean leads to a better performance. This is because prototypes lie on subspaces in the feature space. Additionally, prototypes in the PN can be easily utilized to design a hybrid with our approach.

**Remark 1.** *Interestingly, in special cases, PSN simplifies to PN. For example, if samples of a class span a infinitesimal region, at the limit, collapsing to a point, then an affine subspace reduces to a point, recovering PN from PSN.*

**Remark 2.** *Since it is not possible to build an affine subspace from only one example, PSN cannot address 1-shot learning problems per se. However, simply applying augmentations to support images facilitates 1-shot learning. However, this issue is beyond the point we make in this work.*

**Subspace Dimension.** In comparison to other models such as matching networks or PN, PSN comes with an extra hyper-parameter, the dimensionality of the subspaces (*i.e.*, $n$). As a rule of thumb, we recommend to use $n = K - 1$ to train the model, while $n = 2$ at the test time gives reasonable and robust results. That said, we thoroughly studied the effect of this parameter and summarized our findings in Appendix B. In short, PSN exhibits a great degree of robustness to $n$, which in turns, makes training of them painless.

**Computational Complexity.** The computational complexity of our PSN approach is $\mathcal{O}(\min(ND^2K, NDK^2))$, where $K$, $N$, and $D$ are the number of shot, way, and feature dimensionality respectively. Compared to the complexity of the PN algorithm, *i.e.*, $\mathcal{O}(NDK)$, our algorithm is somehow slower due to the involvement of the SVD step. If the complexity is of a concern, fast approximate algorithms for SVD (*e.g.*, Menon & Elkan (2011)) can be considered.

## 7  CONCLUSIONS

This paper presents the PSN, a novel approach for few-shot learning that employs nonlinear embeddings and modeling via affine subspaces. Empirically, we showed that the representations learned via PSN were expressive across a wide-range of supervised and semi-supervised few-shot problems.

In PSN, each class is represented by the subspace formed by all its examples, meaning that each class is modeled by the span of its samples. One possibility to extend the PSN modeling is to benefit -cleverly- from the null space associated with each class, in the hope of designing a more discriminative model.

---

[2]One can argue why not modeling classes with linear subspaces, *i.e.*, to drop the class mean for modeling. We note that linear subspaces are indeed special cases of more general affine subspaces where all subspaces intersect at the origin.

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

# Appendices

## A  PERTURBATIONS EFFECT ON PERFORMANCE

In this section, we study the effect of perturbation on the performance of PN and PSN. More specifically, we considered the problems of 5-way 5-shot and 5-way 10-shot learning and introduced two types of perturbations at the test time with the trained models. Note that, the model is obtained from 5-way 5-shot training without perturbations. Firstly, we randomly sampled examples from classes not presented in the support sets. Secondly, additive noise is generated randomly using a multivariate Gaussian distribution with random mean and variance of $\sigma = \{0.15, 0.3, 0.4\}$. Both examples in these two types are included in the support examples for prototypes and subspaces creation.

The results of this study are depicted in Fig. 4. To summarize, our experiments show that both PSN and PN are affected by outliers negatively. That said, the PSN exhibits a much better degree of resilience to outliers. For example, modelling with PN leads to a drop of 19% and 12% percentage points when each support set has 20 outliers for the problem of 5-way 5-shot and 5-way 10-shot respectively. For the same experiment, the PSN modelling only suffers 11% and 9% percentage points of performance drop. When additive noise is considered, PSN behaves robustly for a wide-range of contamination. In contrast, the performance of PN drops rapidly and significantly in the presence of noise, reinforcing our idea that subspaces form indeed a more robust model for the task in hand.

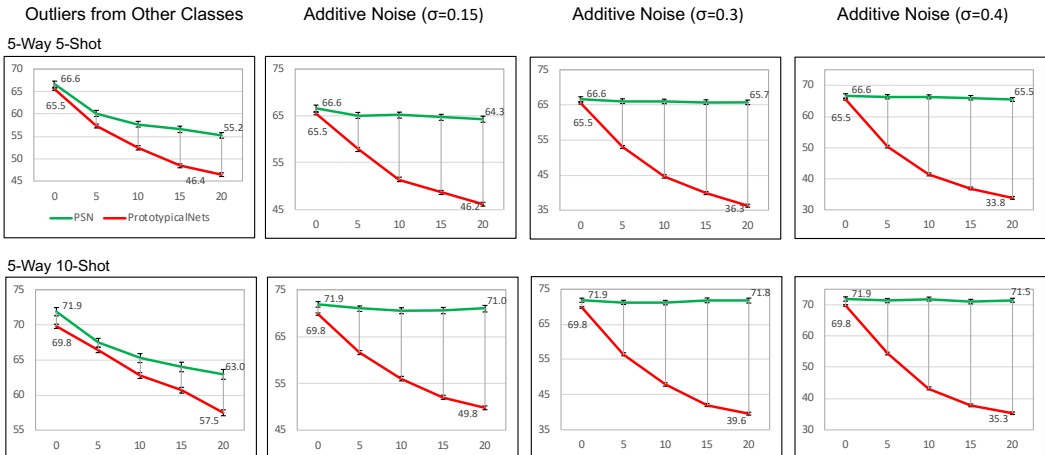

Figure 4: The charts show 5-way 5-shot and 5-way 10-shot results of the PSN and the PN algorithms in the presence of outliers and additive noise. In the first column, sub-plots show the effect of introducing outliers among support samples (the classes of outliers are disjoint with the support classes of samples). The second to fourth columns show the effect of introducing noisy examples generated randomly from Gaussian distributions with random means and variance of $\sigma = \{0.15, 0.3, 0.4\}$, respectively. The performance is measured with increasing number of outliers and noisy examples (X-axes).

## B    SUBSPACE DIMENSION EXPERIMENT

In this section, we report the results of experiments conducted to assess the effect of the subspace dimensionality (*i.e.*, $n$) on the overall accuracy of the algorithm. In particular, we considered two few-shot problems, namely 5-way 5-shot and 5-way 20-shot. For the former, we varied the subspace dimensionality $n \in \{2, 3, 4, 5\}$ and considered all combinations during training and testing (for example, $n = 4$ during training and $n = 3$ at the test time). For the experiment on 5-way 20-shot problem, we varied the subspace dimensionality $n \in \{5, 10, 15, 20\}$ and again considered all combinations.

Our experiments demonstrate that PSN behaves robustly over a wide-range of subspace dimensionality. For example, the lowest and highest performances for the 5-way 5-shot learning are $66.08 \pm 0.67\%$ and $66.62 \pm 0.69\%$, respectively. For the problem of 5-way 20-shot learning, the lowest and highest performances are $75.03 \pm 0.57\%$ and $75.58 \pm 0.57\%$, respectively.

