# OpenReview forum: "Projective Subspace Networks For Few-Shot Learning"
_ICLR.cc/2019/Conference_

### Official Review · AnonReviewer1 · 2018-10-23
**Neat idea but requires clarifying the merits of the approach and differences from previous work**

**Rating:** 6
**Confidence:** 4

**Review:**

This paper presents a new method for fully- and semi-supervised few-shot classification that is based on learning a general embedding as usual, and then learning a sub-space of it for each class. A query point is then classified as the class whose sub-space is closest to it.

Pros: This is a neat idea and achieves competitive results. Learning a sub-space per class makes intuitive sense to me since it’s plausible that there is a lower-dimensional subspace of the overall embedding space that captures the properties that are common to only examples of a certain class. If this is indeed the case, it seems that indeed classifying query examples into classes based on their distances from the corresponding sub-spaces would lead to good discrimination.

Cons: First, an inherent limitation is that this approach is not applicable to one-shot learning, and I have doubts in its merit for very low shot learning (explained below). Second, I’m missing the justification behind a key point used to motivate the approach, which requires clarification (explained below). Third, I feel that certain aspects of the approach were unclear (details to follow). Finally, I feel more analysis is needed to better understand the differences of this method from previous work (concrete suggestions follow). For semi-supervised learning, the novelty regarding how the unlabeled examples are incorporated is limited, as the approach used is previously-introduced in Ren et al, 2018.

Overall, even though I like the idea and the results are good, there are a few points, mentioned in the above section that I feel require additional work before I can strongly recommend acceptance. Most importantly, relating to getting more intuition about why and when this works best, and tying it in better with previous approaches.

A key point requiring clarification.
There is a key fact that the authors used to motivate this approach which remains unclear to me: why is it the case that this approach is less sensitive to outliers than previous approaches? In Figure 1, an outlier is pictured in each of subfigures (a) and (b) corresponding to Matching and Prototypical Networks, but not in subfigure (c) which corresponds to PSN. No explanation is provided to justify this conjecture, other than empirical evaluation that is based on the overall accuracy only. In particular, since SVD is used to obtain the sub-spaces, instead of an end-to-end learned projector that directly optimizes the query set accuracy, it’s not clear why if a support point is an outlier it would not affect the sub-space creation. If I’m missing something, please clarify!

(A) Comments on the approach.
(1) Why define X_k as the support set examples minus the class prototype instead of just the support examples themselves? The latter seems simpler, and should have all the required information for shaping the class’ subspace.
(2) Note that if X_k is defined as [x_{k,1}, \dots, x_{k,K}] as proposed in the above point (ie. without subtracting the class mean from each support point) then this method would have been applicable to 1-shot too. How would it then compare to a 1-shot Prototypical Network? Notice that in this case the mean of the class is equal to this one example.
(3) In general, the truncated SVD decomposition for a class can be written using the matrices U, \Sigma and V^T with dimensions [D, n], [n, n] and [n, K] respectively, where D is the embedding dimensionality and K is the number of support points belonging to the given class. The middle matrix \Sigma in the non-truncated version would have dimensions [D, K]. Does this mean that when truncating, n is enforced to be smaller than each of D and K? This would mean that the dimensionality n of the sub-space is limited by the number of the support examples, which in some cases may be very small in few-shot learning. Can you comment on this?
(4) How to set n (the dimensionality of each subspace) is not obvious. What values were explored? Is there a sweet spot in the trade-off between the observed complexity and the final accuracy?

(B) Comparison with Prototypical Networks.
(1) In what situations do we expect learning a sub-space per class to do better than learning a  prototype per class? For example, Figure 4 shows the test-time performance as a function of the test ‘way’. A perhaps more interesting analysis would be to compare the models’ performance as a function of the test *shot*: if more examples are available it may be less appropriate to create a prototype and more beneficial to create a sub-space?
(2) Can we recover Prototypical Networks as a special case of PSN? If so, how? It would be neat to show under which conditions these are equivalent.

(C) Clarifications regarding the semi-supervised setup.
(1) Are distractor classes sampled from a disjoint pool of classes, or is it that, for example, a class which is a distractor in an episode is a non-distractor in another episode.
(2) Similarly for labeled / unlabaled at training time. Can the same example appear as labeled in one episode but unlabaled in another? In Ren et al, 2018, this was prevented by creating an additional labeled/unlabeled split even for the training examples. Therefore they use strictly less overall information at meta-training time than if that split weren’t used. To be comparable with them, it’s important to apply this same setup.

(D) Additional minor comments.
(1) “To work at the presence of distractors, we propose to use a fake class with zero mean”. Note that this was already proposed in Ren et al, 2018. They used a zero-mean, high-variance additional cluster whose aim was to ‘soak up’ the distractor examples to prevent them for polluting legitimate clusters (this was the second model they proposed).
(2) In the introduction, regarding contribution iii. A more appropriate way to describe this is as exploring generalization to different numbers of classes, or ‘ways’ at test time than what was used at training time.
(3) Gidaris and Komodakis (2018) is described in the related work as using a more complicated pipeline. Note however that their pipeline is in place for solving a more challenging problem than standard few-shot classification: they study how a model can maintain the ability to remember training classes while rapidly learning about new ‘test’ classes.
(4) In the last line of section 5.3, use N-way instead of K-way since in the rest of the paper K was used to refer to the shot, not the way.

---

> ### Author Response · Authors · 2018-11-26
> **Response to AnonReviewer1- Part 2**
>
>
> Q: ``How to set n (the dimensionality of each subspace) is not obvious.''
>
> A: To address the concern here, we performed an extra experiment on various subspace dimensions ($n$) (kindly see Appendix B for the details).
> In particular, we considered the problems of 5-way 5-shot and 5-way 20-shot and studied the effect of subspace dimensionality on overall performances. For instance, we varied $n$ from 2 to 5 in both training and testing for the 5-way 5-shot experiment. The experiments show that the algorithm is robust to the dimensionality to a great degree. Our rule of thumb and recommendation in the paper is to set the dimensionality of subspace as $K-1$ for training and $2$ for testing (kindly see Appendix B).
>
> Q: ``if more examples are available it may be less appropriate to create a prototype and more beneficial to create a sub-space?''
>
> A: For this comment, we have provided a thorough analysis by increasing the value of ways and shots in Table 3 of our revised work. In this experiment, we trained
> PSN and PN on 5-way 5-shot setting. Then, at the test time, we provided the trained models with support-sets of various ways and shots.
> This indeed simulates more realistic tests. Our proposed method can outperform PN with a tangible gap when the number of ways and shots increases.
>
> Q: Can we recover Prototypical Networks as a special case of PSN? If so, how? It would be neat to show under which conditions these are equivalent.
>
> A: If the embedded features have a very minuscule variance, PSN is equivalent to PN. We really appreciate this comment and have reflected it in Section 6.
>
> Q: (a)Are distractor classes sampled from a disjoint pool of classes, or is it that, for example, a class which is a distractor in an episode is a non-distractor in another episode.
> (b)Similarly for labeled / unlabaled at training time. Can the same example appear as labeled in one episode but unlabaled in another?
>
> A: There is a disjoint pool between unlabeled and labeled examples applied to all classes. Unlabeled data in an episode contains only examples from unlabeled pool. Moreover, there are two types of experiments conducted for semi-supervised learning: with and without distractor classes. The distractor class is taken from other classes irrelevant to the current episode and items are sampled from unlabeled pool. Note that a distractor class can appear in another episode as a non-distractor class (this is the protocol).
>
> Q: (a) ``In the introduction, regarding contribution iii. A more appropriate way to describe this is as exploring generalization to different numbers of classes, or ‘ways’ at test time than what was used at training time.''
> (b) ``In the last line of section 5.3, use N-way instead of K-way''
>
> A: Thank you, we have revised the relevant text based on the above comment.

---

> ### Author Response · Authors · 2018-11-26
> **Response to AnonReviewer1 - Part 1**
>
>
> Q: ``Why is it the case that ... less sensitive to outliers ...''
>
> A: A prototype in the Prototypical Networks (PN) is the average of a set and as such is sensitive to any perturbation of the set, outliers being one. On the other hand, in PSN, a set is represented by a subspace. To have a noticeable change in the orientation of the subspace, one needs to induce drastic changes to the set. Having said this, we performed two extra-experiments to reinforce our conjecture here.
>
> In the first one, depicted in Fig. 2, we empirically studied the decision boundaries of PSN and PN. The samples (triangle symbol) are drawn from the normal distribution for a two-class problem (column 1 and 2) and a problem with three classes (column 3 and 4). The outliers (square symbol) are spread around initial samples. The facecolors for outliers indicate to which class they have been assigned. In the odd columns, we can see that the prototypes and subspaces discriminate the classes equally well.
> However, in the even columns, it is clearly shown that the outliers sway the prototypes and their decision boundaries while the subspace approach handles them more robustly.
>
> In the second experiment, placed in appendix A, we provide the 5-way 5-shot and 5-way 10-shot results on the Mini-ImageNet by adding outliers and noise to support examples. There are two setups conducted to examine the robustness of PSN to outliers and additive noise. In the first experiment, the support set contains samples from classes absent in this set. We did not split the dataset into disjoint inlier/outlier sets though, as the outliers were only presented at the test time, leading to more realistic experiments. In the second experiment, perturbations were generated from a Gaussian distribution with random mean and predefined
> variance. The results shown in Fig. 4 demonstrate that on both tasks, PSN outperforms PN by a significant gap. Note that, we utilized the same CNN architecture (4-convolutional layers) for both PSN and PN.
>
> Q: ``Why define $X_k$ as the support set examples minus the class prototype instead of just the support examples themselves?''
>
> A: Our idea here is to represent a class by an affine subspace which is indeed a generalization of
> the concept of linear subspace (where the origin is a common point). We indeed started by using linear-subspaces to model each class but empirically found that affine subspaces perform slightly better. To address this comment, we have added a remark to Section 6.
>
> Q: ``How would it then compare to a 1-shot Prototypical Network?''
>
> A: We cannot build an affine subspace with only one example per class, as such we cannot use PSN to address 1-shot learning problems per se. However, simply augmenting support images to obtain two or more samples per class alleviates such an issue straight away. However, this simple issue is beyond the points we make in our paper (it is an orthogonal research direction in one- and few-shot learning). We have reflected this in Section 6 of the revised draft to address the reviewer's comment.
>
> Q: ``Does this mean the dimensionality n of the sub-space is limited by the number of the support examples''
>
> A: Affirmative. Our idea is to construct a subspace representing the set. The reviewer's comment raises an interesting point, whether parts of the orthogonal complement of each subspace can be used to make better decisions. We believe that investigating the effect of orthogonal complements demands a dedicated study and goes beyond our work. Having said this, we reflect this comment in Section 7.

---

### Official Review · AnonReviewer3 · 2018-11-03
**Some experiments are missing.**

**Rating:** 6
**Confidence:** 3

**Review:**

This paper proposes a Projective Subspace Network (PSN) for few-shot learning. The PSN represents each support set of classes as a subspace obtained by SVD. Then the method calculates distances between a query and classes by the projection error to the subspace. Instead of using the prototype of the class center, the subspace representation is more robust to outliers. Though the contribution seems to be incremental, it is a reasonable improvement upon Matching Networks and Prototypical Networks.

Pros.
+ The proposed subspace method is simple and reasonable
+ The performance is better than some related works on few-shot learning.

Cos.
- The authors claimed that subspace representation is more robust to noise within each class samples. However, this is not supported by experiments. The authors evaluated the distractor classes. However, this is not the case when the outlier existed within each class samples.

- For semi-supervised few-shot learning, the authors proposed a fake class with zero means. The effect of this fake class is not evaluated.

- The dimensionality of subspace (n) seems to be not written.

- The sensitivity analysis of the dimensionality of subspace is missing. For subspace methods, it is essential to evaluate the performance w.r.t the dimension.

- Descriptions in the related work section should be improved. It is unclear how the proposed method is related to K-means, K-modes, and K-prototype. Also, the authors wrote that works (Chan et. 2015, Sun et al. 2017) use PCA or SVD to reduce the dimensionality of feature representation in neural networks. However, both methods do not perform dimensional reduction. PCANet (Chan et al . 2015) obtains convolutional filters by applying PCA to input images or feature maps. SVDNet (Sun et al. 2017) applies SVD for obtaining decorrelated weights in a neural network.

---

> ### Author Response · Authors · 2018-11-26
> **Response to AnonReviewer3**
>
>
> Q:  The authors claimed that subspace representation is more robust to noise within each class samples. However, this is not supported by experiments.
>
> A: To address this comment, we have revised our paper and have added a new figure along new experiments to study the effect of outliers. Kindly see our response to AnonReviewer2 which we repeat below for convenience:
>
> We have added a new figure (Fig.2) to our paper to address your comment. There, we empirically studied the decision boundaries of PSN and PN. The samples (triangle symbols) are drawn from the normal distribution with two (column 1 and 2) and three( column 3 and 4) different classes. The outliers (square symbols) are spread around initial samples. The facecolors of the outliers show to which class the outliers are assigned to.
>
> In addition, we empirically studied the effect of outliers on the Mini-ImageNet dataset (see Section 5.3 and Appendix A). Therein, we considered two types of perturbations on two few-shot learning problems, namely  5-way 5-shot and 5-way 10-shot. In all the aforementioned experiments, we observed that PSN is more robust to outliers compared to PN.
>
> Q: The dimensionality of subspace (n) seems to be not written.  The sensitivity analysis of the dimensionality of subspace is missing.
>
> A: We have revised our work (section 6) and added
> an appendix (Appendix B) to our paper and studied the sensitivity of PSN to the dimensionality of subspaces. Based on our experiments, PSN is robust to variations in their subspace dimensionality to a great degree.
>
> Q: For semi-supervised few-shot learning, the authors proposed a fake class with zero means. The effect of this fake class is not evaluated.
>
> A: We actually 'borrow' this idea from Ren et al. (2018) (fake class with zero mean) but we adapt it to our subspace model. However, unlike the variant of PN for semi-supervised learning proposed by Ren et al.(2018), our approach does not need to model any subspaces for the distractor classes.
>
> Q: Descriptions in the related work section should be improved.
>
> A: We have taken this advice on board and revised Section 2 accordingly. Aside from rewording, we have added references to methods that use backpropagation through SVD.

---

> > ### Comment · AnonReviewer3 · 2018-11-30
> > **Thank you for the revision.**
> >
> > The authors have added the visualization examples of robustness to outliers/noise in class samples of the training set, and experimental results to show its effectiveness. I think this result is fine.
> >
> > They also have clarified the dimensionality of subspace and the less sensitivity to the subspace dimensionality. However, the authors only reported the highest and lowest performances. Thus, it is unclear why the subspace $n=K-1$ for training while $n=2$ for testing gives reasonable and robust results.
> >
> > For the fake class, I wanted to see the performance without this fake class in semi-supervised learning. If the authors only borrowed the idea of Ren et al. (2018), it should be clarified in Sec.4.2.
> >
> > I think the concerns of robustness to noise and subspace dimensionality have been improved in some degrees. So, I would like to change my recommendation to marginal accept.
> >
> > Minor problem.
> > Ramark1 in the revised manuscript describes that when a class samples span an infinitesimal region, then affine subspace becomes to a point. However, SVD on data with small variation would produce orthogonal bases U and small singular value matrix \Sigma. Thus, using U, the affine subspace does not become a point. I agree with the case when we truncate the subspace corresponding to small singular values. In such case, all bases in U will not be used, i.e. it becomes in the case n=0.

---

> > > ### Author Response · Authors · 2018-12-07
> > > **Comment**
> > >
> > > We want to thank the reviewer for his/her insightful suggestions and support given in preparations of our revised manuscript. Of course, we agree with the Remark 1 and we have already rectified this in our off-line draft. We are more than happy to take into account any further suggestions the reviewer may have for our work.

---

### Official Review · AnonReviewer2 · 2018-11-04
**Interesting extension to embedding-based approaches to few-shot learning but results are a bit disappointing**

**Rating:** 6
**Confidence:** 4

**Review:**

This paper considers the problem of few-shot learning and proposes a new embedding-based approach. In contrast to previous work (such as Matching Networks and Prototypical Networks) where distance is computed in pure embedding space, this work proposes computing a low-dimensional subspace to represent a class and using the distance from an embedded query point to this subspace. The low-dimensional subspace for a class is computed by running truncated Singular Value Decomposition on the normalized embeddings of all points in the support set for that class and using the top n left singular vectors as the basis for the class's subspace. The authors also propose an extension to their model to the semi-supervised few-shot learning setting by incorporating masked-mean computation and zero-mean cluster for distractor items (both ideas borrowed from Renn 2017 for prototypical networks). Experiments are conducted on Mini-Imagenet in the few-shot learning setting and on Mini-Imagenet and Tiered-ImageNet in the semi-supervised few-shot learning setting.

Pros:
- Proposed idea is novel and proposes an interesting change to existing embedding-based few-shot learning techniques.

Cons:
- Performance benefit is a bit disappointing; Mini-Imagenet few-shot performance improvement relative to Prototypical-Nets is minimal (barely 1% for 5way-5shot and 20way-5shot case). For semi-supervised experiments, there is bigger improvement for Mini-Imagenet (4% for both without distractors and with distractors) but less so for Tiered-ImageNet (close to 0% for without distractors and with distractors).

Remarks:
- The paper seems to be missing what the dimensionality of the subspace is for the experiments? Was this picked using validation set performance?
- In first paragraph of page 2, it seems too strong to say "...this makes our paper unparalleled to previous studies"; maybe change to "...this make our proposed model novel relative to previous work"
- Is there previous work that has involved back-propagating through SVD? It would be useful to mention these as references.
- In Figure 1, it is visually shown how outliers can negatively impact Matching-Networks and Prototypical-Networks but not visually shown how PSN is resistant to them?
- The claim is made that the proposed method is more robust to outliers. Is there more of a justification that can be given for this? Either in terms of some intuition or an experiment that can be run? For example, can it be shown that outliers cause the prototype of a class to move a lot (in terms of distance from original prototype without outliers) whereas the original subspace compared to subspace with outliers is less different by measuring this on Mini-Imagenet?
- Typo on page 5: "in what follwos" => "in what follows"
- In Discussion, paper states, "Moreover, the Prototypical Network makes use of the class mean and can be easily incorporated in our testbed": what does this mean exactly?

---

> ### Author Response · Authors · 2018-11-26
> **Response to AnonReviewer2**
>
>
> Q: ``The paper seems to be missing what the dimensionality of the subspace is for the experiments?''
>
> A: In our initial submission, we used a rule of thumb to set the dimensionality (n is $K$-1 during training and two at the testing time).
> To address the reviewer's comment, we studied the effect of subspace dimension in Section 6 and Appendix B. We examined on four different subspace dimensions on 5-way 5-shot and 5-way 20-shot for training and testing stage and found that the performance is not affected badly.
>
> Q: it seems too strong to say "...this makes our paper unparalleled to previous studies"
>
> A: Duly noted. We have rephrased the introduction according to your comment.
>
> Q: ``Is there previous work that has involved back-propagating through SVD?''
>
> A: Yes, in particular the work of  Ionescu et al. [1], Li et al. [2], and Gou et al. [3] used backpropagation through SVD to address semantic segmentation, large scale classification, and visual recognition problems. We have revised section 2 to introduce these works.
>
> [1] Catalin Ionescu, Orestis Vantzos, and Cristian Sminchisescu. Matrix backpropagation for deep networks with structured layers. ICCV, 2015.
>
> [2] Peihua Li, Jiangtao Xie, Qilong Wang, and Wangmeng Zuo. Is second-order information helpful forlarge-scale visual recognition?. ICCV, 2017.
>
> [3] Mengran Gou, Fei Xiong, Octavia Camps, and Mario Sznaier. MoNet: Moments Embedding Network. CVPR, 2018.
>
> Q: ``In Figure 1, ... but not visually shown how PSN is resistant to them?''
>
> A:   We have added a new figure (Fig. 2) to our paper to address this comment. There, we empirically studied the decision boundaries of PSN and PN. The samples (triangle symbols) are drawn from normal distribution with two (column 1 and 2) and three (column 3 and 4) different classes. The outliers (square symbols) are spread around initial samples. The facecolors of the outliers show to which class the outliers are assigned to.
>
> In addition, we empirically studied the effect of outliers on the Mini-ImageNet dataset (kindly see Section 5.3 and Appendix A). Therein, we considered two types of perturbations on two few-shot learning problems, namely 5-way 5-shot and 5-way 10-shot. In all the aforementioned experiments, we observed that PSN is more robust to outliers as compared to PN.
>
> Q: In Discussion, paper states, "Moreover, the Prototypical Network makes use of the class mean and can be easily incorporated in our testbed": what does this mean exactly?
>
> A: We meant that it was possible to design a hybrid model and benefit from the inference mechanism provided by the prototypes along our PSN. However, since our aim in this paper is to show and contrast the advantage of modelling with subspaces, we did not pursue such a development. Thus, we removed the confusing text from our revised paper.
>
> Q: ``Performance benefit is a bit disappointing''
>
> A: Our experiments show that PSN is superior to Prototypical Networks (PN) in all cases.
> More importantly, per our new experiments added to the new draft, PSN is more robust to outliers. Furthermore, PSN copes better with the increasing number of classes (ways) and makes a better use of unlabeled samples (as shown in Section 5).  We believe this shows that PSN is a far better model for the problem at hand.

---

> > ### Comment · AnonReviewer2 · 2018-12-05
> > **Thank you for the revision**
> >
> > The experiments studying the effect of outliers for the Mini-Imagenet dataset are very interesting and are an effective way to display that PSN is indeed more robust against different types of outliers. Additionally, thank you for clarifying how the dimensionality of subspace impacts results.

---

> > > ### Author Response · Authors · 2018-12-07
> > > **Comment**
> > >
> > > We want to thank the reviewer for his/her insightful input and suggestions that led to an improved version of our paper. We have taken all remarks on-board when making our revision. It is our pleasure if the reviewer found our revisions/additional experiments interesting and valuable. We hope other readers will also enjoy our work. We are more than happy to take into account any further suggestions the reviewer may have for our work.

---

### Author Response · Authors · 2018-11-26
**General Comments**

We thank the reviewers for their valuable comments and feedback.
We have taken onboard all the comments and revised our work accordingly. In particular,

1. We empirically studied the robustness of our algorithm to outliers for the Mini-ImageNet dataset.
This is reflected in Section 5 and Appendix A of our revised draft.
Our study reveals that compared to Prototypical Networks (PN hereafter), PSN is more robust to outliers and additive noise.

2. We  studied the effect of the dimensionality of the subspaces in PSN. This extra experiment is discussed in Section 6 and Appendix B. Our experiments show that PSN behaves robustly with respect to subspace dimensionality in a wide-range.

3. We also studied a form of generalization ability of PSN in Section 5. This is achieved by increasing the number of shot (K) and way (N) for the experiment on the Mini-ImageNet. Again, we observed that PSN outperforms PN by a visible margin.


Below, we provide point-by-point responses to comments by quoting excerpts from the reviews.

---

### Meta-Review · Area_Chair1 · 2018-12-13
**A robust extension of prototypical networks, but needs a clear motivation for this property.**

**Confidence:** 3
**Recommendation:** Reject

**Metareview:**

The reviewers all like the idea, and though the performance is a little better when compared to prototypical networks, the reviewers felt that the contribution over and above prototypical networks was marginal and none of them was willing to champion the paper. There is merit in that there is increased robustness to outliers, and future iterations of the paper should work to strengthen this aspect.

As a quick nitpick: based on my reading, and on Figure 3, it looks like there might be a typo in the definition of X_k (bottom of page 4). Right now it is defined in terms of the original data space x, when I think it should be defined in terms of the embedding space f(x). Overall this paper is a good contribution to the few-shot learning area.